# A highly tunable dopaminergic oscillator generates ultradian rhythms of behavioral arousal

Ian D Blum[1,2,4], Lei Zhu[1,2], Luc Moquin[2], Maia V Kokoeva[3], Alain Gratton[1,2], Bruno Giros[1,2,5], Kai-Florian Storch[1,2]*

[1]Department of Psychiatry, McGill University, Montreal, Canada; [2]Douglas Mental Health University Institute, Montreal, Canada; [3]Department of Medicine, McGill University, Montreal, Canada; [4]Integrated Program in Neuroscience, McGill University, Montreal, Canada; [5]INSERM UMR S1130 CNRS, UMR8246, Sorbonne University, Paris, France

**Abstract** Ultradian (~4 hr) rhythms in locomotor activity that do not depend on the master circadian pacemaker in the suprachiasmatic nucleus have been observed across mammalian species, however, the underlying mechanisms driving these rhythms are unknown. We show that disruption of the dopamine transporter gene lengthens the period of ultradian locomotor rhythms in mice. Period lengthening also results from chemogenetic activation of midbrain dopamine neurons and psychostimulant treatment, while the antipsychotic haloperidol has the opposite effect. We further reveal that striatal dopamine levels fluctuate in synchrony with ultradian activity cycles and that dopaminergic tone strongly predicts ultradian period. Our data indicate that an arousal regulating, dopaminergic ultradian oscillator (DUO) operates in the mammalian brain, which normally cycles in harmony with the circadian clock, but can desynchronize when dopamine tone is elevated, thereby producing aberrant patterns of arousal which are strikingly similar to perturbed sleep-wake cycles comorbid with psychopathology.

*For correspondence: florian. storch@mcgill.ca

Competing interests: The authors declare that no competing interests exist.

## Introduction

Ultradian rhythms with periods ranging from one to several hours have been linked to various aspects of mammalian physiology. Usually superimposed on the 24-hr diurnal or circadian rhythm, ultradian oscillations have been observed in the context of locomotion, sleep, feeding, body temperature, serum hormones, and brain monoamines in species ranging from fruit flies to humans (*Tannenbaum and Martin, 1976*; *Ibuka et al., 1977*; *Rusak, 1977*; *Daan and Slopsema, 1978*; *Honma and Hiroshige, 1978*; *Dowse et al., 1987*; *Rivkees, 2003*; *van Oort et al., 2007*; *Dowse et al., 2010*; *Seki and Tanimura, 2014*). These physiological cycles most frequently exhibit periods of 2–6 hr, adopting harmonics of the 24-hr daily light–dark cycle or the endogenous circadian rhythm, when external timing cues are absent. However, the generation of such ultradian rhythms does not depend on a functional circadian system nor a light:dark cycle. Ultradian locomotor oscillations persist in rodents housed in constant darkness even upon ablation of the suprachiasmatic nucleus (SCN), the site of the master circadian pacemaker (*Ibuka et al., 1977*; *Rusak, 1977*), or genetic disruption of the circadian clock (*Vitaterna et al., 1994*; *Bunger et al., 2000*) (*Figure 1A,B*). These ultradian activity cycles may not simply be driven by metabolic demand since the 2- to 3-hr rhythm in foraging activity observed in the common vole persists even in the absence of food (*Gerkema and van der Leest, 1991*). Studies in this species further indicate that one adaptive value of ultradian activity rhythms may lie in the facilitation of social synchrony, which is suggested to reduce predator risk in this species

**eLife digest** The sleep-wake cycle of mammals is controlled by a 'circadian clock' within the brain, which is synchronized to the day–night cycle. However, other aspects of mammalian physiology including alertness and activity levels, as well as appetite and body temperature—fluctuate in cycles that repeat every few hours. These cycles are known as ultradian rhythms, and they may offer survival benefits by enabling potentially risky behaviors, such as foraging, to be coordinated between members of a group.

Despite their widespread nature and the fact that they appear to be conserved in evolution, virtually nothing is known about the molecular basis of ultradian rhythms. Blum et al. have now identified a second internal clock within the brain, which they name 'the DUO', and shown that this clock normally works in concert with the circadian clock to regulate daily patterns of activity and alertness.

Experiments in mice revealed that the DUO uses the brain chemical dopamine to generate bursts of activity roughly every four hours. Moreover, it continues to work when the circadian clock has been destroyed. Measurements of dopamine in freely moving mice showed that levels of the chemical fluctuate in synchrony with the animals' activity levels. Moreover, drugs that flood the brain with dopamine, such as methamphetamine, disrupt the 4-hour cycle by lengthening the period between bursts of activity, whereas drugs that block dopamine receptors have the opposite effect.

As well as revealing a mechanism by which the brain coordinates processes that repeat several times per day, the identification of the DUO could also provide insights into the biological basis of psychiatric disorders. Conditions such as schizophrenia and bipolar disorder are often accompanied by disturbances in patterns of activity and rest. While these have previously been attributed to the disruption of circadian rhythms, there is little direct evidence for this, which raises the possibility that these changes might instead reflect the disruption of ultradian rhythms.

(*Gerkema and Verhulst, 1990*). However, despite their prevalence and hypothesized biological significance, ultradian locomotor rhythms have received little research attention, likely owing to their frequently masked expression and unstable periodicity in contrast with circadian activity rhythms (*Ruis et al., 1987*; *Schibler, 2008*) (*Figure 1C,D*).

In addition to ultradian oscillations, methamphetamine-induced rhythms of locomotor activity occur in the absence of a functional circadian clock. When provided in the drinking water, methamphetamines (Meth) induce a daily activity bout in addition to the expected circadian rhythm (*Honma et al., 1986*). This Meth-dependent component—which typically adopts a period in the circadian range—is not abolished upon SCN lesion nor by genetic disruption of circadian clock function (*Honma et al., 1987*; *Mohawk et al., 2009*). It was thus concluded that a methamphetamine-sensitive circadian oscillator (MASCO) outside the SCN exists which is capable of driving daily cycles of locomotor activity (*Tataroglu et al., 2006*). Despite the longstanding recognition of ultradian and Meth-dependent rhythms, the underlying cellular and molecular identity of the oscillator(s) driving them is unknown.

Here, we provide evidence for a highly tunable dopaminergic ultradian oscillator (DUO) which is continuously operative in the mammalian brain and which, together with the circadian clock, orchestrates the daily pattern of arousal. Our data suggest that dopamine acts as both the principal oscillator output as well as an integral component of the DUO, determining oscillator period. Our findings further indicate that the previously described MASCO represents a long-period manifestation of the DUO resulting from elevated dopamine tone. Importantly, our data support an intriguing proposition: that DUO, rather than circadian clock, dysregulation critically contributes to the sleep-wake abnormalities associated with psychopathology.

## Results

### Dopamine transporter deficiency results in ultradian locomotor period lengthening

To gain insights into the mechanistic basis of ultradian locomotor rhythm generation, we considered that locomotor activity is associated with an awakened state (*Welsh et al., 1988*) and consequently,

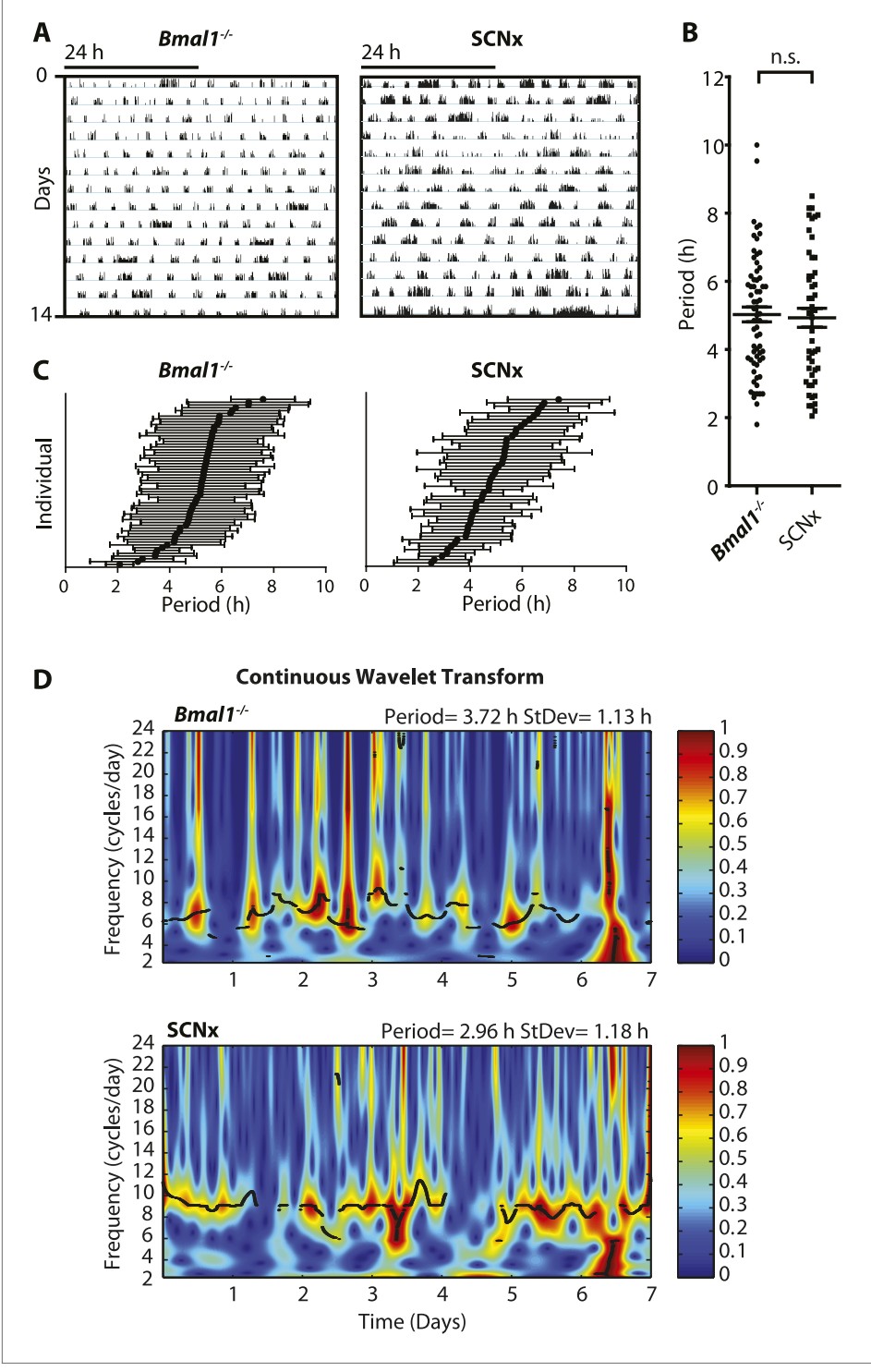

**Figure 1**. Inter- and intra-animal variability of ultradian activity rhythms across time in circadian incompetent mice. (**A**) Representative double-plotted actograms of running wheel activity in SCN-lesioned (SCNx) and *Bmal1⁻/⁻* mice in DD. (**B**) Dot plot of locomotor period length in DD based on Lomb-Scargle periodogram analysis of seven consecutive days of activity recording (N = 65 for *Bmal1⁻/⁻* and N = 48 for SCNx; $t_{111}$ = 0.2785, p = 0.78, unpaired t-test). (**C**) Intra-animal period variability expressed as mean ± SD for each animal, ranked according to mean period length derived from continuous wavelet transforms (CWT) for the 1–12-hr frequency range (same animals and timespans for calculation as in **B**). (**D**) CWT-heatmaps showing decibel scaled and normalized amplitude of oscillations according to frequency and time with black traces indicating the ridge of local amplitude maxima.

the ultradian locomotor rhythms observed in mice that lack circadian clock function (*Figure 1*) could be interpreted as rhythms of heightened wakefulness or arousal. In mammals, a key role in arousal promotion has been attributed to distinct monoaminergic neuronal populations located in the upper brainstem and midbrain (*Brown et al., 2012*). While altering extracellular levels of the arousal-associated monoamines serotonin, norepinephrine, or histamine by genetic manipulation has only limited effects on locomotion (*Thomas and Palmiter, 1997*; *Bengel et al., 1998*; *Xu et al., 2000*; *Parmentier et al., 2002*; *Zhao et al., 2006*), depleting the brain of dopamine (DA) profoundly abrogates locomotor activity (*Zhou and Palmiter, 1995*). Moreover, increasing extracellular DA levels induces hyperlocomotion (*Giros et al., 1996*) and lengthens the time spent awake (*Wisor et al., 2001*). We therefore speculated that altering DA tone may affect ultradian rhythm generation. To test this, we examined running wheel activity in mice carrying a disruption in the *Slc6a3* gene, which encodes the dopamine transporter (DAT). *Slc6a3*$^{-/-}$ mice exhibit hyperdopaminergia due to the lack of DAT-mediated DA reuptake into dopaminergic neurons, leading to a hyperactivity phenotype (*Giros et al., 1996*; *Gainetdinov et al., 1998*). As the presence of the circadian clock and/or a light:dark cycle frequently masks ultradian activity rhythms (*Schibler, 2008*), we assessed the locomotor behavioral consequences of DAT elimination in the absence of the master SCN circadian pacemaker. To do so, we electrolytically lesioned the SCN of *Slc6a3*$^{-/-}$ mice and their wildtype littermates and monitored their running wheel behavior in constant darkness (DD). While control mice (SCNx-*Slc6a3*$^{+/+}$) exhibited ultradian activity rhythms with the expected ~4-hr period, SCNx-*Slc6a3*$^{-/-}$ mice showed rhythms whose periods were three times longer (*Figure 2A,B*). Analysis of mice that were deficient for both DAT and the essential clock component BMAL1 (*Bmal1*$^{-/-}$, *Slc6a3*$^{-/-}$) corroborated this finding. *Bmal1*$^{-/-}$, *Slc6a3*$^{-/-}$ mice exhibited ~12–14-hr rhythms in locomotor activity, largely phenocopying the SCNx-*Slc6a3*$^{-/-}$ mice, while their *Bmal1*$^{-/-}$, *Slc6a3*$^{+/+}$ littermates showed ~4-hr periods as expected for isodopaminergic mice lacking circadian clock function (*Figure 2C,D*). Together, these results suggest that DAT removal markedly increases ultradian cycle length. Alternatively, the ~12-hr rhythms observed in SCNx-*Slc6a3*$^{-/-}$ and *Bmal1*$^{-/-}$, *Slc6a3*$^{-/-}$ animals may originate from an independent oscillator, one that is activated by DAT elimination, while the short period ultradian oscillator that operates in DAT intact, SCNx *or Bmal1*$^{-/-}$ animals is disengaged or otherwise obscured.

## The psychostimulant methamphetamine lengthens ultradian locomotor period

In order to corroborate that DAT removal indeed lengthens the period of the ultradian activity cycles, we took into account that DAT-mediated DA uptake can be reversed by the selective action of the psychostimulant methamphetamine (Meth) (*Howell and Kimmel, 2008*). Because Meth leads to increased extracellular DA concentrations as in the case of *Slc6a3* gene disruption, we speculated that Meth treatment would result in a, possibly gradual, period lengthening of the ultradian locomotor rhythms. Indeed when we treated *Bmal1*$^{-/-}$ animals with increasing concentrations of Meth via drinking water in DD, we observed a gradual lengthening of the initial ~4-hr locomotor oscillations and this was accompanied by a corresponding increase in activity bout length (*Figure 3A,B*). Of note, the period increase in response to elevating Meth concentrations did not halt in the circadian range, rather, oscillations continued to lengthen with individual animals reaching periods of 100 hr or more (*Figure 3C*). Gradual period lengthening of ultradian locomotor rhythms was also observed in *Bmal1*$^{-/-}$ animals exposed to amphetamine (*Figure 3—figure supplement 1A,B*), a drug similarly targeting DAT (*Howell and Kimmel, 2008*), but with lower efficacy than Meth (*Goodwin et al., 2009*). These results argue that DAT targeting psychostimulants affect an endogenous ultradian rhythm generator by increasing period length. However, due to the mode of delivery (drinking water) and the rhythmic Meth uptake that may in turn result from it, it is conceivable that the Meth-dependent, long-period oscillations are 'driven' by rhythmic drug intake rather than being generated endogenously. To address this possibility, we subcutaneously implanted *Bmal1*$^{-/-}$ mice with osmotic minipumps that continuously infused Meth over a period of 2 weeks. Running wheel analysis in DD of Meth-infused *Bmal1*$^{-/-}$ animals demonstrated a significant ultradian period lengthening upon drug infusion (*Figure 3—figure supplement 1C,D*) suggesting that rhythmic uptake is not required for Meth to exert its period lengthening effect, in line with a previous study performed in rats (*Honma et al., 1987*). The relatively limited change in period observed in this experimental paradigm could be due to the short, 2-week infusion timespan, which is perhaps insufficient to robustly lengthen periods beyond 12 hr (*Figure 3—figure supplement 1D*). Together, these findings support the notion that the long-period oscillations observed

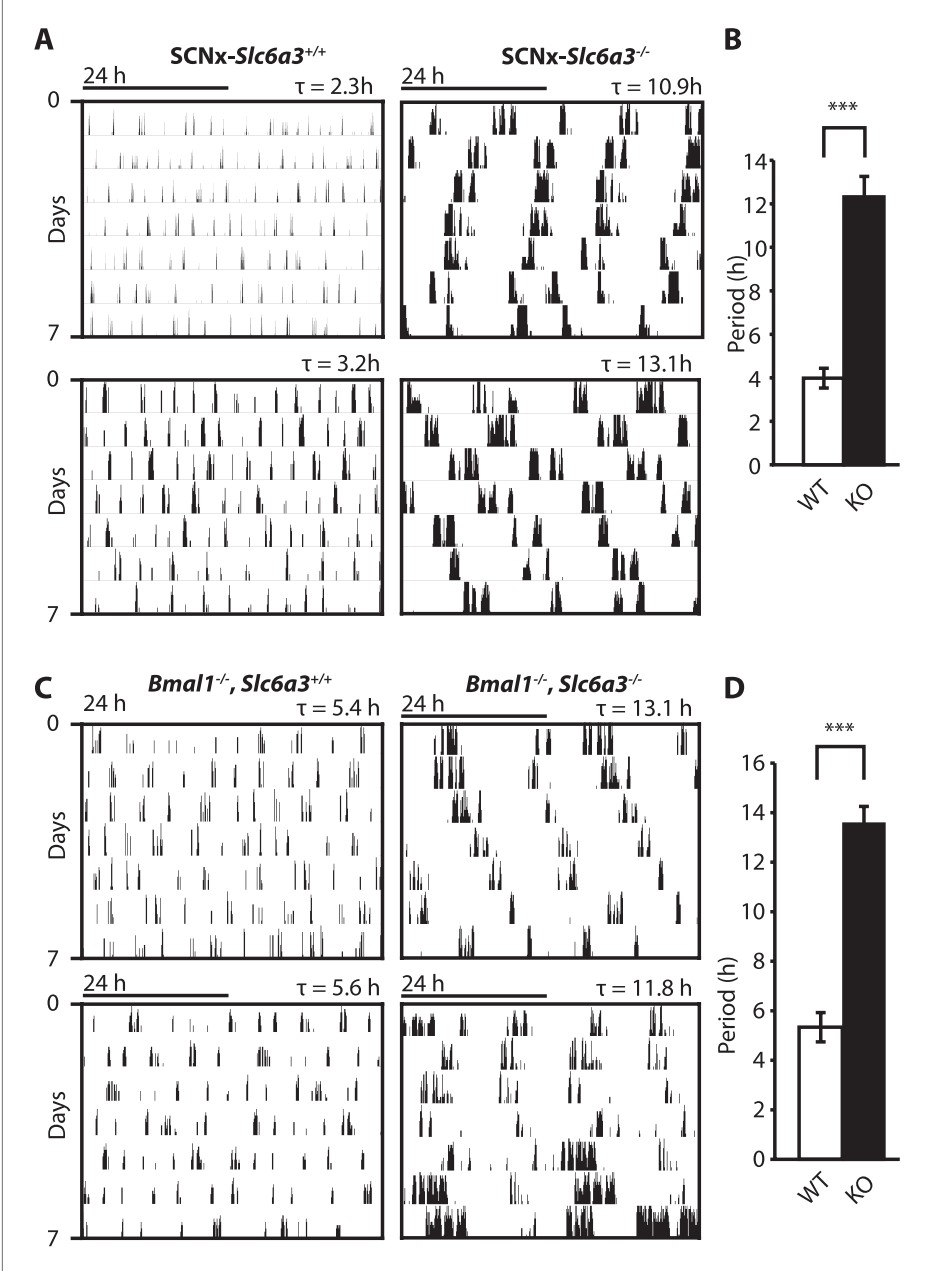

**Figure 2**. Dopamine transporter knockout alters periodicity of ultradian locomotor rhythms in circadian incompetent mice under constant darkness. (**A**) Representative, double-plotted actograms demonstrating marked lengthening of ultradian locomotor periods in SCNx-*Slc6a3*$^{-/-}$ mice as compared to SCNx-*Slc6a3*$^{+/+}$ littermates. Tau(τ) indicates individual period. (**B**) Period length averages of ultradian activity (N = 7; $F_{1,6}$ = 253.8, ***p < 0.0001, ANOVA) from Lomb-Scargle periodogram analysis of 7-day time-spans. (**C**) Representative, double-plotted actograms demonstrating markedly increased ultradian period lengths in *Bmal1*$^{-/-}$, *Slc6a3*$^{-/-}$ mice as compared to *Bmal1*$^{-/-}$, *Slc6a3*$^{+/+}$ mice. (**D**) Period length averages of ultradian activity (N = 4; $F_{1,3}$ = 194.2, ***p < 0.001, ANOVA) from Lomb-Scargle periodogram analysis of 7-day time- spans.

in Meth-treated animals, and likewise in *Slc6a3*$^{-/-}$ mice, are due to period expansion of an endogenous ultradian rhythm of arousal.

We next aimed to confirm these findings in intact mice. While activity rhythms in the ultradian range are easily discernable in voles and hamsters (*Gerkema et al., 1990*; *van der Veen et al., 2006*; *Prendergast et al., 2012*), the presence of such rhythms in mice or rats is much less obvious and this

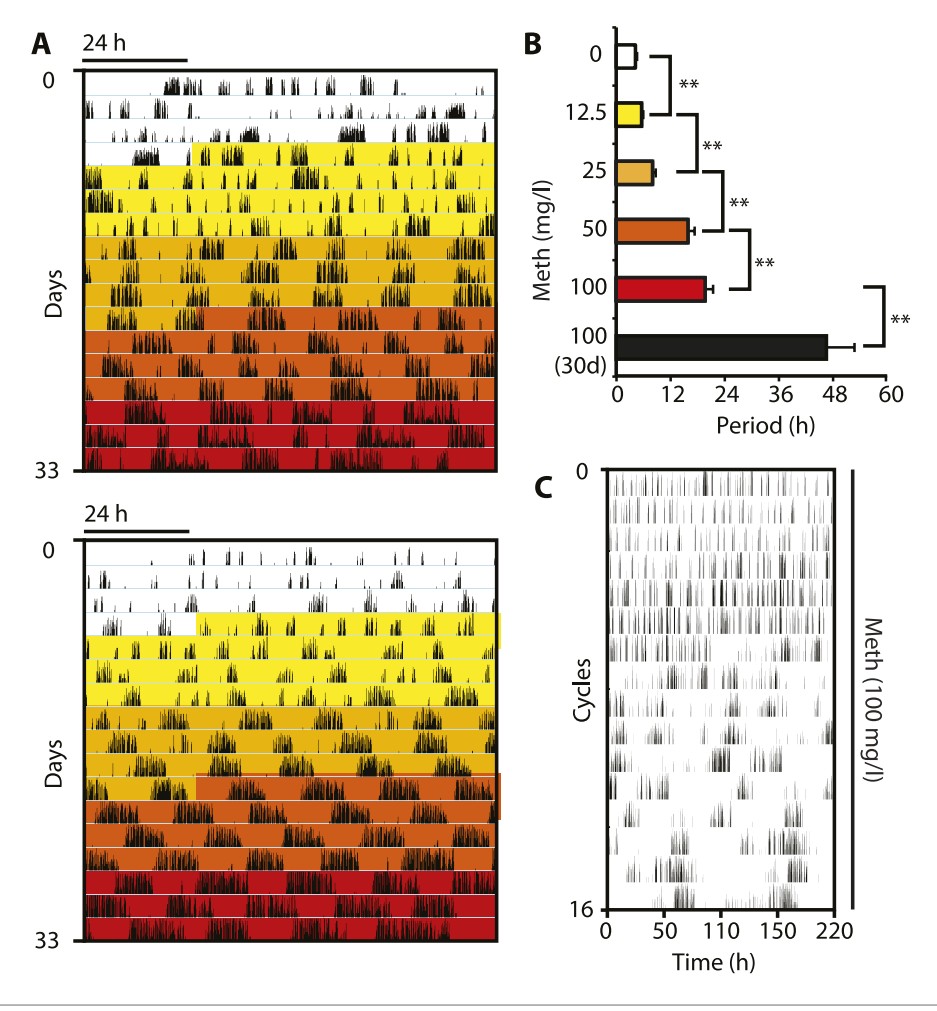

**Figure 3**. Pharmacological tuning of DAT activity by incrementally increasing methamphetamine concentrations lengthens ultradian locomotor period into the infradian range. (**A**) Representative actograms of Meth-treated *Bmal1*−/− mice in DD. Treatment intervals are highlighted with corresponding concentrations indicated in (**B**). (**B**) Mean periods from the last 7 days at a given Meth concentration. Repeated measures ANOVA revealed a significant main effect of concentration/time ($F_{5,40}$ = 34.30, p < 0.001) and significant period lengthening between consecutive concentrations (mean ± SEM; N = 9; **p ≤ 0.005, planned comparisons; 30 day, 30 day exposure to 100 mg/l). (**C**) Modulo 110-hr actogram of a *Bmal1*−/− animal after extended exposure to Meth revealing an ultra-long activity rhythm.

The following figure supplement is available for figure 3:

**Figure supplement 1**. Amphetamine treatment lengthens ultradian locomotor rhythms in *Bmal1*−/− mice.

has been attributed to masking by the circadian clock (*Schibler, 2008*). Indeed, running wheel activity data from intact mice often do not provide strong indications of ultradian activity rhythms. However, when we recorded ambulatory behavior using telemetry, we frequently detected three, evenly spaced activity bouts during the active (night) phase (*Figure 4A*). The observed ~4 hr peak-to-peak spacing is in line with a previous study on several circadian competent mouse strains where ultradian rhythms with similar period length were detected (*Dowse et al., 2010*). The absence of a clearly discernable ~4-hr rhythmic component during the light portion of the daily LD cycle (*Figure 4A*) is likely due to masking by the SCN and/or light. Unlike WT mice, *Slc6a3*−/− mice never showed a triple-peak activity pattern at night (*Figure 4B*), and spectral density analysis in the ultradian (2–8 hr) range (*Figure 4C*) underscores that rhythm generation is significantly altered in these animals while total daily activity is not (*Figure 4D*). When we provided C57BL/6 mice carrying a telemetry implant with Meth in their

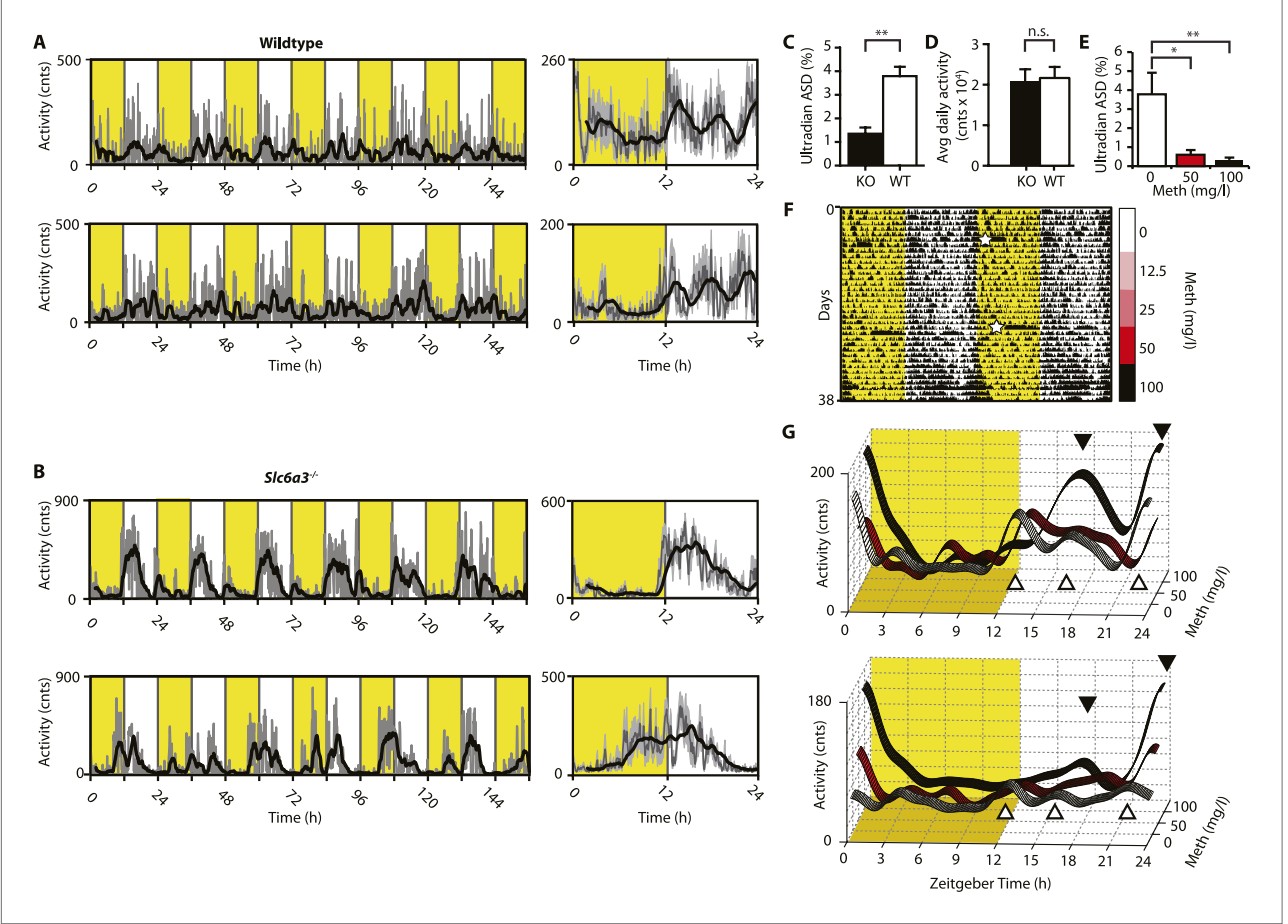

**Figure 4**. Ultradian activity in SCN-intact *Slc6a3*[−/−] mice and their wildtype littermates. (**A** and **B**) Ambulatory activity recorded by telemetry implants across multiple days (left) and averaged over 24 hr (right). Traces represent 2-hr recursive smoothing (black) of the underlying raw DATa (dark grey; SEM envelope, light grey). Areas in yellow indicate lights on. (**C**) Amplitude spectral density in the ultradian range (2–8 hr) is significantly different between *Slc6a3*[−/−] mice and their wildtype littermates revealing a loss of the ultradian component (mean ± SEM; N = 10; $F_{1,18}$ = 26.40, **p < 0.0001, ANOVA). (**D**) Although the temporal pattern of locomotion differs between genotypes there is no significant difference in daily activity averaged over multiple days (mean ± SEM; N = 10, $F_{1,18}$ = 0.1793, ANOVA). (**E–G**) Addition of Meth to the drinking water of C57BL/6 mice lengthens the night-time activity bouts in a concentration-dependent manner. Averaged daily locomotor activity of individual mice at different Meth concentrations (**G**) derived from the time-span indicated by colored bars next to the representative actogram (**F**) and subjected to Butterworth filtering. The three night-time activity peaks before treatment (white triangles), transform to 2 peaks after exposure to the highest concentration (black triangles). The night-time bout lengthening is also reflected in the reduction of ultradian amplitude spectral density in the 1 to 5-hr range (**E**, mean ± SEM, N = 7; $F_{2,20}$ = 8.08, p < 0.005, repeated measures ANOVA; *p < 0.05, **p < 0.01, post-hoc Bonferroni). White asterisks (in **E**) indicate cage changes.

drinking water, we observed a gradual lengthening of the interval between the night-time activity peaks (***Figure 4F,G***), resulting in the transformation of the triple peak (***Figure 4G***, white trace and white triangles) into a dual or single night-time activity peak pattern at the highest (100 mg/l) Meth concentration (***Figure 4G***, black trace and black triangles). This was corroborated by spectral density analysis revealing a marked reduction in ultradian frequencies upon Meth treatment (***Figure 4E***). Together, these results indicate that blocking DA reuptake also lengthens ultradian activity bout intervals in circadian competent mice, arguing that an ultradian oscillator is not 'activated' by circadian disruption, but rather continuously operative alongside the circadian timer. As with *Bmal1*[−/−] mice, prolonged Meth-treatment in intact WT mice can lead to profound period lengthening of ultradian rhythms, often into the circabidian (48 hr) range (for examples see Figures 5E, 7F, 9B).

## The antipsychotic haloperidol shortens ultradian locomotor period
Given that both DAT disruption and Meth treatment elevate extracellular DA and concurrently lengthen ultradian rhythm period, we speculated that manipulations aimed at lowering DA tone

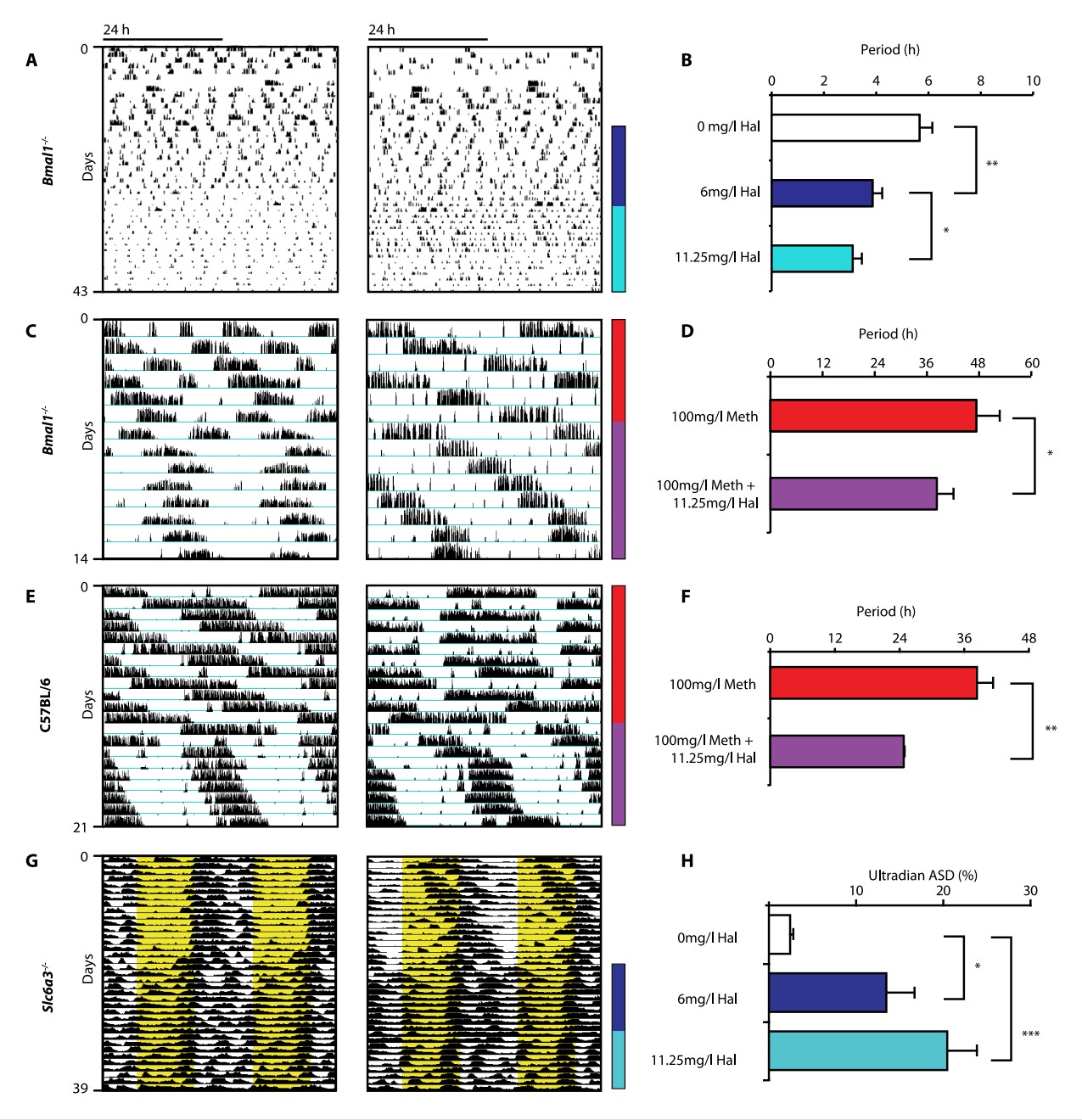

**Figure 5**. Haloperidol shortens circadian-clock-independent locomotor rhythms. Representative actograms with Hal treatment periods indicated by colored bars (left); bar graphs indicate corresponding locomotor period (right). (**A**) Increasing concentrations of Hal provided in the drinking water gradually shortens the endogenous ultradian activity rhythms of $Bmal1^{-/-}$ mice in DD (**B**, mean ± SEM, N = 12; $F_{2,10}$ = 14.36, p < 0.0001, repeated measures ANOVA; *p < 0.05 and **p < 0.01, planned comparison ANOVA). (**C** and **E**) Hal shortens the infradian rhythms in Meth-treated $Bmal1^{-/-}$ mice (**D**, mean ± SEM, N = 9; $F_{1,8}$ = 2.357, *p < 0.05 ANOVA) and WT mice (**F**, mean ± SEM, N = 9; $F_{1,8}$ = 3.525, **p < 0.01, ANOVA) in DD. (**G**) Hal treatment increases the frequency of temperature fluctuation in $Slc6a3^{-/-}$ mice under LD measured by telemetric implants. (**H**) Changes of amplitude spectral density in the ultradian range (2–8 hr) in response to increasing Hal concentrations (mean ± SEM; N = 8; $F_{2,7}$ = 14.74, repeated measures ANOVA, *p < 0.05, ***p < 0.0005, post-hoc Bonferroni). For all experiments, periods are calculated based on the running wheel activity during the last 7 days of treatment at the indicated Meth/Hal concentration.

should conversely lead to period shortening. We therefore provided $Bmal1^{-/-}$ mice in DD with the antipsychotic drug haloperidol (Hal) in their drinking water. Hal selectively blocks the DA receptor D2 (DRD2) (**Strange, 2008**), which is expressed on postsynaptic target sites of DA neuronal projections

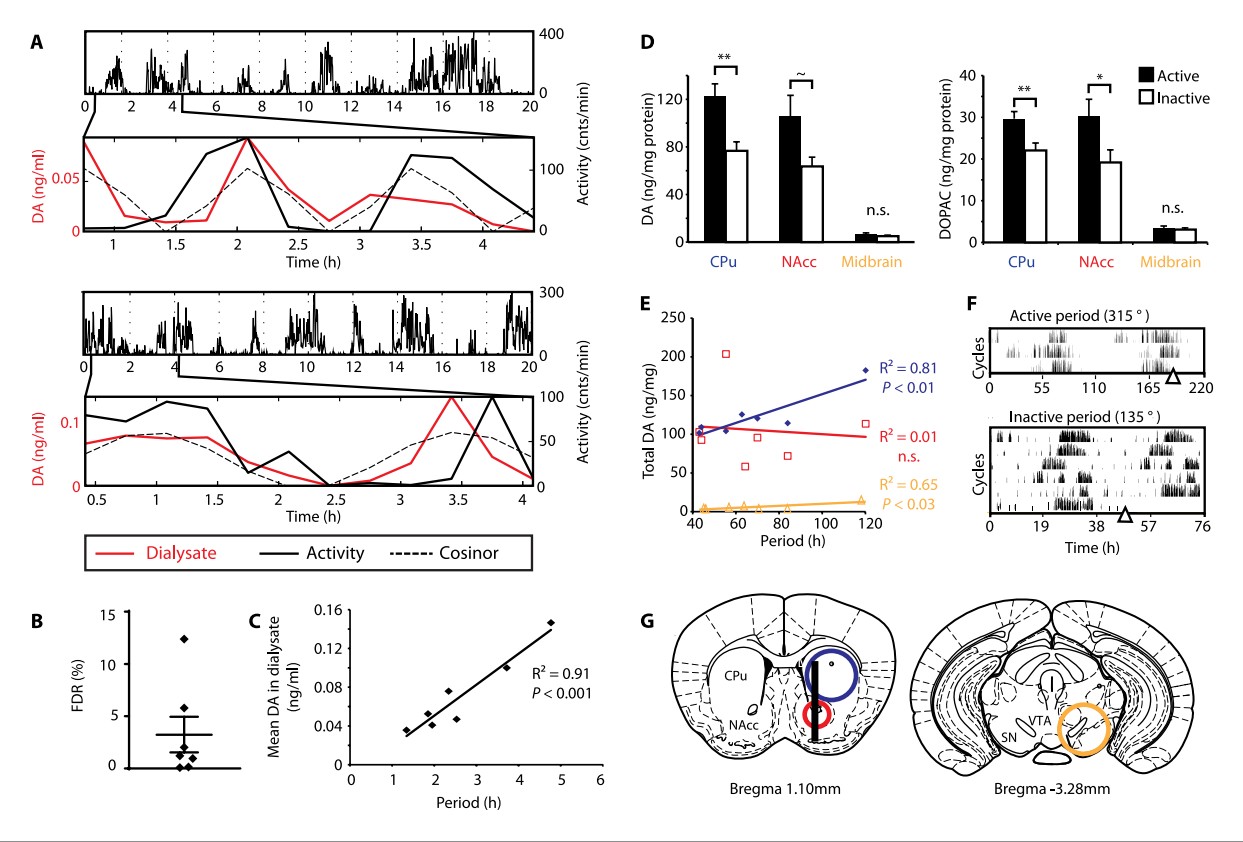

**Figure 6**. DA fluctuations correlate with ultradian locomotor behavior in *Bmal1$^{-/-}$* mice. (**A**) Two representative examples of in vivo striatal microdialysis in *Bmal1$^{-/-}$* mice. Upper: locomotor activity as measured by beam breaks. Lower: DA dialysate concentration measured at 20-min intervals (red trace) plotted alongside the corresponding locomotor activity (solid black trace). Cosinor (dotted line) was computed from the 20 hr of locomotor activity following dialysate sampling. (**B**) False discovery rate of the fit between the DA profiles and corresponding cosinors (mean ± SEM, N = 7). (**C**) Linear regression analysis of period length vs mean DA concentration, dots representing individual animals. (**D**) Tissue punches of CPu, NAcc, and midbrain were analyzed for DA and DOPAC content in animals sacrificed during their active (■) vs their non-active (□) phase (mean ± SEM; active phase, N = 7, inactive phase, N = 6; *p < 0.05, **p < 0.01, ~p = 0.066, ANOVA). (**E**) Linear regression for period vs DA content in animals sacrificed during the active phase revealed significant correlations for the CPu and midbrain. (**F**) Representative actograms displaying activity rhythms of Meth-treated *Bmal1$^{-/-}$* mice used for tissue collection. Triangles indicate collection time points (in °), with locomotor activity bout onset set to 180°. (**G**) Illustrations of coronal mouse brain sections (*Paxinos and Franklin, 2001*) indicating position of the active membrane (black bar) and tissue punch placements (circles, colors correspond to labels in **D** and **E**).

The following figure supplement is available for figure 6:

**Figure supplement 1**. Rhythms of extracellular DA in the striatum of freely-moving *Bmal1$^{-/-}$* mice.

but also presynaptically as an autoreceptor (*Schmitz et al., 2003*). Importantly, when given chronically as in our case, Hal has been reported to electrically silence midbrain DA neurons (*White and Wang, 1983*) and to markedly lower extracellular DA levels in the striatum/nucleus accumbens regions in rats (*Lane and Blaha, 1987*; *Ichikawa and Meltzer, 1991*). As predicted, *Bmal1$^{-/-}$* mice responded with successive locomotor period shortening to increasing concentrations of Hal (*Figure 5A,B*). We also observed Hal-mediated shortening of the long-period behavioral rhythms in *Bmal1$^{-/-}$* and wild-type mice treated concurrently with Meth (*Figure 5C–F*). A period shortening effect of Hal could also be discerned from the core body temperature fluctuations of *Slc6a3$^{-/-}$* mice, which increased in frequency (*Figure 5G*). This response, which mirrored the locomotor behavioral response (not shown), was concentration dependent and spectral density analysis confirmed a successive increase of the ultradian component upon Hal exposure (*Figure 5H*). Together, these data strongly support the hypothesis that both the ultradian rhythms observed in wildtype and *Bmal1$^{-/-}$* mice and the long-period rhythms previously attributed to the MASCO are driven by the exact same oscillatory mechanism.

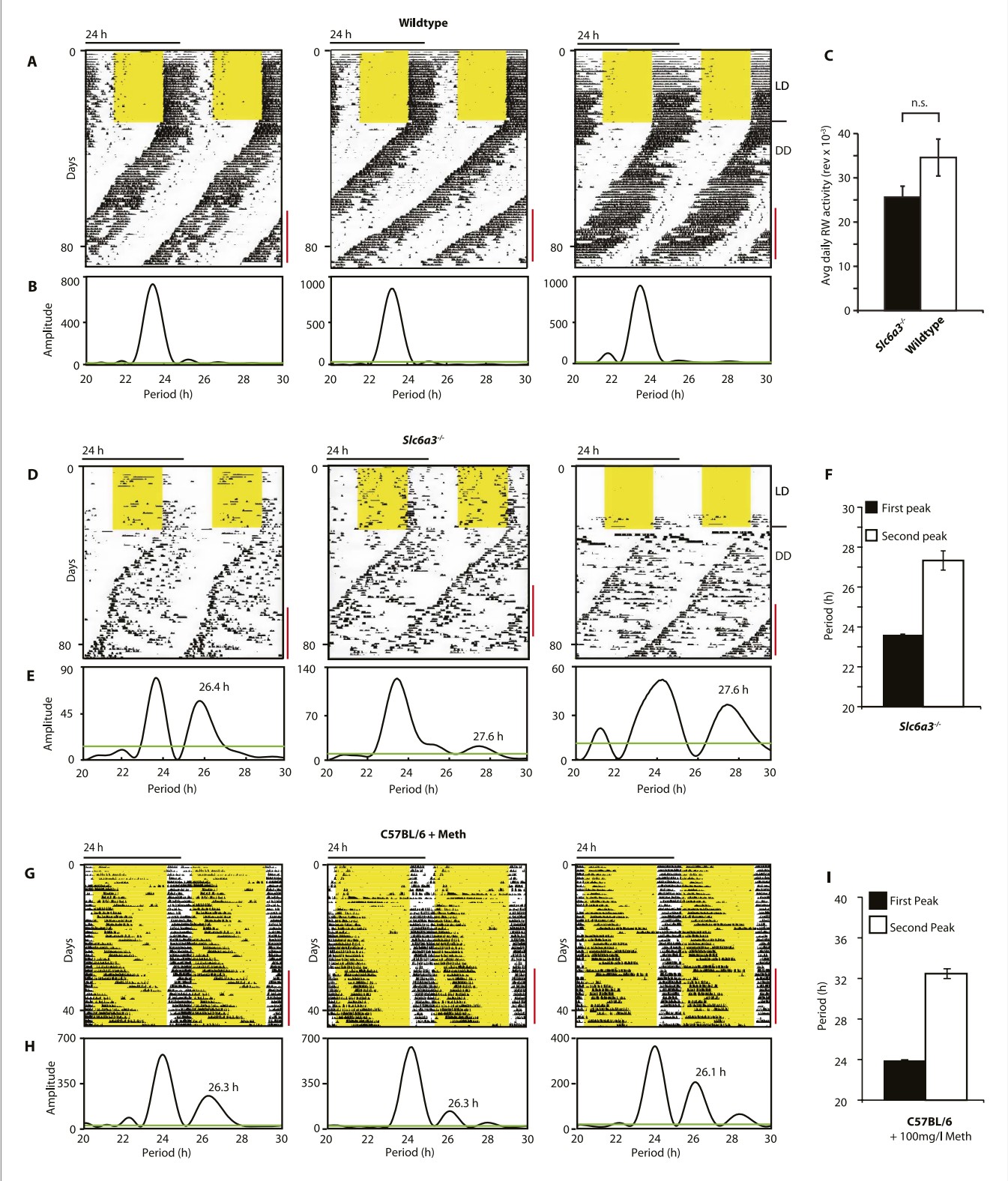

**Figure 7**. *Slc6a3*[−/−] mice show a second rhythmic locomotor activity component. Representative actograms displaying daily running wheel activity of *Slc6a3*[+/+] (**A**), *Slc6a3*[−/−] mice (**D**) and C57BL/6 mice on Meth (**G**). (**B**, **E** and **H**) Lomb-Scargle periodograms generated from the time-span indicated by red bars, in (**A**, **C**, **G**), respectively. (**C**) There is no significant difference between genotypes in daily activity averaged over the time-span of analysis (mean ± SEM; N = 6, $F_{1,10}$ = 1.848, ANOVA) (**F** and **I**) Average periods of the highest 2 periodogram peaks for *Slc6a3*[−/−] mice (**E**, mean ± SEM, N = 6)
*Figure 7. Continued on next page*

*Figure 7. Continued*

and C57BL/6 mice on Meth (**H**, mean ± SEM, N = 9, Meth-treatment started on day 1 of the recordings). Areas in yellow indicate lights-on. Green line in the periodograms indicates the confidence threshold for rhythmicity ($\alpha$ = 0.01).

The following figure supplement is available for figure 7:

**Figure supplement 1**. The SCN of *Slc6a3*$^{-/-}$ mice is unperturbed.

As altering extracellular DA is a common denominator of all three manipulations—DAT elimination, (meth) amphetamine, and haloperidol treatment—these findings collectively suggest that DA tone determines ultradian period. Given that DA is known to mediate arousal/wakefulness (*Brown et al., 2012*), it appears plausible that DA also serves as principal oscillator output, driving rhythms in arousal. We thus hypothesized that extracellular DA must oscillate in synchrony with the observed activity rhythms.

## Extracellular DA fluctuates in synchrony with ultradian activity cycles

To test this hypothesis, *Bmal1*$^{-/-}$ mice were unilaterally implanted with a microdialysis probe positioned along the ventro-dorsal extent of the right striatum (*Figure 6G*), a site heavily innervated by DA neurons. Mice were kept under constant dim red light (<5lux) and ultradian activity rhythms were monitored using infra-red beam breaking. The animals showed the expected short period rhythms in locomotor activity throughout the experiment (*Figure 6A*), and analysis of dialysates revealed that extracellular DA levels fluctuated concordantly with the activity cycles (*Figure 6A* and *Figure 6—figure supplement 1*, compare solid red and black traces). Importantly, we generated a model oscillation using the cosinor method from locomotor rhythms recorded for 20 hr after dialysate sampling with no investigator present and compared it to the DA fluctuations observed. The goodness of fit between the cosinor and the DA profiles indicated by the sum of squared errors (SSE = 0.050 ± 0.027, mean ± SEM, N = 7) suggests that the dialysate sampling procedure itself did not perturb the generation of the endogenous ultradian activity cycles. To determine the statistical significance of the observed agreement between the cosinor and the DA trace, we randomly permuted the individual time points of the DA trace and determined the percentage of permuted traces with an equal or better fit to the cosinor in comparison to the observed DA fluctuation. On average, only 3.2 ± 1.7% of 100,000 permuted DA profiles correlated as well or better than the experimentally observed DA measurements (*Figure 6B*), confirming that extracellular DA fluctuates in synchrony with the ultradian activity cycles. Of particular note, linear regression revealed a highly significant correlation between mean DA concentration in the dialysate and ultradian locomotor period in the *Bmal1*$^{-/-}$ animals we tested (*Figure 6C*), which again supports a role of (extracellular) DA as a period determinant of the ultradian activity cycle.

As our findings suggest that the observed >24-hr activity rhythms are due to period lengthening of the ultradian oscillations, we similarly expected DA levels to fluctuate in synchrony with the activity cycles in Meth-treated animals. To test this, we measured DA content in tissue punches of Meth-treated *Bmal1*$^{-/-}$ mice from the dorsal (caudate putamen, CPu) and ventral (nucleus accumbens, NAcc) striatum as well as the ventral midbrain, which includes both the substantia nigra (SN) and the ventral tegmental area (VTA) (*Figure 6G*). For punch collection, animals were sacrificed during their active and inactive phases, respectively (*Figure 6F*). We detected significantly higher levels of DA as well as its immediate metabolite, 3,4-dihydroxyphenylacetic acid (DOPAC), in tissue extracts of the CPu during the animal's active phase (*Figure 6D*). There was also a significant increase in DOPAC levels and a trend towards elevated DA (p = 0.066) during the active period in the NAcc, while extracts from the midbrain region, which contains the cell bodies of the striatal dopaminergic afferents, exhibited no detectable change (*Figure 6D*).

We also found a significant correlation between locomotor period and DA content in CPu and midbrain punches collected during the active phase (*Figure 6E*), again consistent with a role for DA as a period determinant of ultradian oscillations.

## SCN-intact *Slc6a3*$^{-/-}$ mice show two components of rhythmic activity

As *Slc6a3*$^{-/-}$ mice lack the triple peak night-time activity pattern that is characteristic of wild-type mice, we subjected *Slc6a3*$^{-/-}$ mice to long-term running wheel activity monitoring to examine their

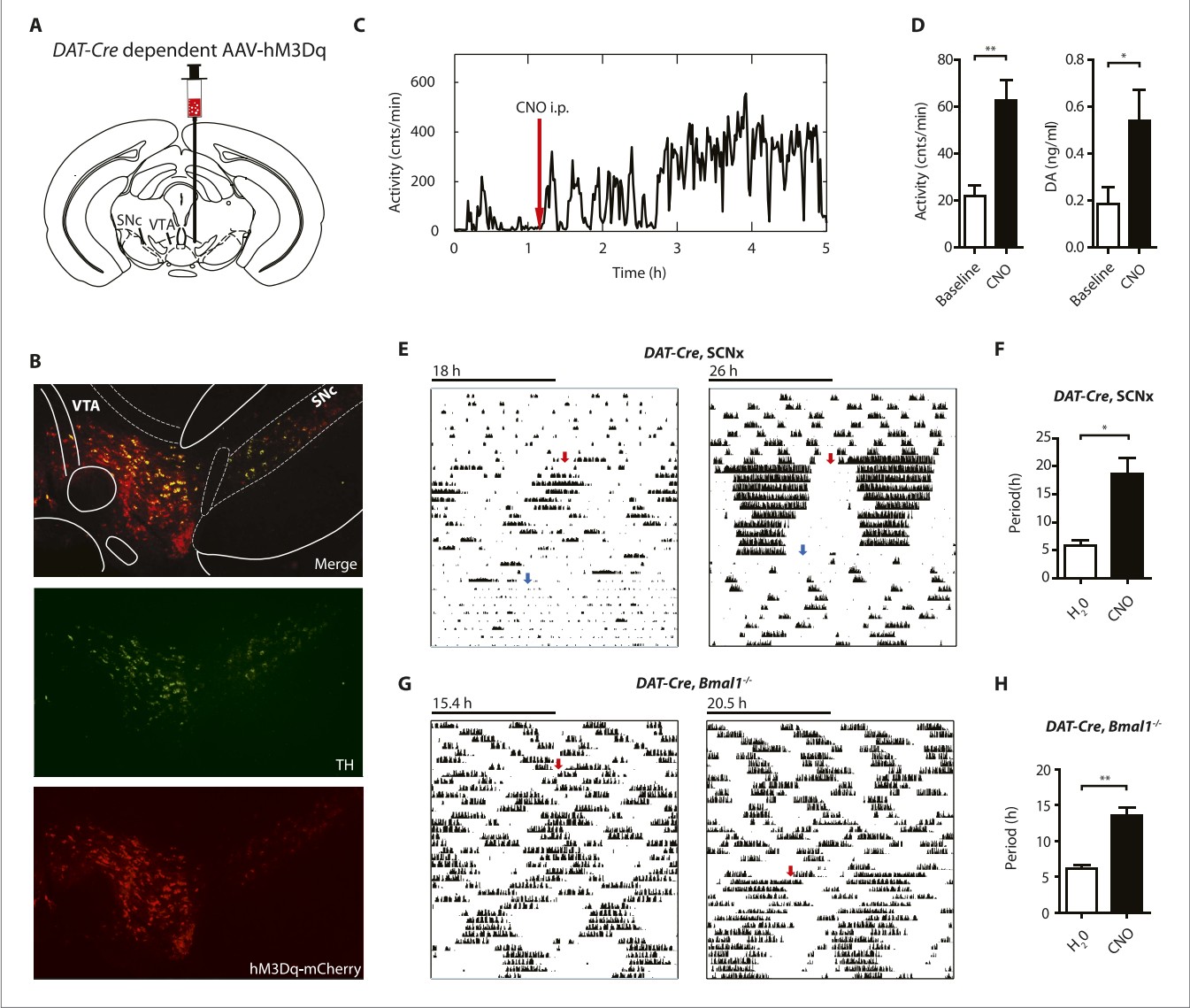

**Figure 8**. Chemogenetic activation of midbrain DA neurons lengthens ultradian locomotor period. (**A**) *AAV-DIO-hM3Dq-mCherry* was stereotaxically and bilaterally delivered into the VTA/SN region of *DAT-Cre* transgenic mice as indicated. (**B**) Representative immuno-fluorescent image of the ventral midbrain from a virus-injected and behaviorally responsive mouse showing extensive co-expression of the mCherry fusion-tag in TH-positive cells of the midbrain. (**C**) Locomotor response to CNO (red arrow, 1 mg/kg body weight i.p.) of a representative, *AAV-hM3Dq* transduced *DAT-Cre* mouse undergoing microdialysis. (**D**) 20min- binned locomotor activity and extracellular striatal DA content (20 min) of *AAV-hM3Dq* transduced mice 1 hr prior (Baseline) and 2 hr after CNO injection. Mice were implanted with a striatal microdialysis probe and DA content was measured as in ***Figure 6A*** (mean ± SEM; N = 3, *p < 0.05, **p < 0.01, paired t-test). (**E–H**) Representative actograms of *AAV-hM3Dq* transduced *DAT-Cre,* SCNx (**E**) and *DAT-Cre, Bmal1⁻/⁻* mice (**G**). Switch to CNO-containing drinking water (7.5 mg/l) is indicated by red arrow. Blue arrow marks return to regular water (in **E**). Animal activity is plotted modulo according to the period measured during the last 7 days of CNO treatment. Periodogram analysis reveals CNO-dependent period lengthening in both *DAT-Cre,* SCNx (**F**, mean ± SEM; N = 6, $F_{1,5}$ = 3.68,*p < 0.05, ANOVA) and *DAT-Cre, Bmal1⁻/⁻* (**H**, mean ± SEM; N = 11; $F_{1,10}$ = 20.48, **p < 0.0001, ANOVA) mice. VTA, ventral tegmental area; SNc, substantia nigra pars compacta.

The following figure supplement is available for figure 8:

**Figure supplement 1**. *AAV-h3MDq* targeting of DA neurons.

locomotor behavior in more detail. Under LD conditions, we observed more fragmented daily activity in *Slc6a3⁻/⁻* than their wildtype counterpars (***Figure 7A,D***). Upon release into constant darkness (DD), wild-type animals exhibited a daily rhythm with a period slightly shorter than 24 hr as expected for mice (***Bunger et al., 2000***) (***Figure 7A***). While this principal locomotor component was also observed

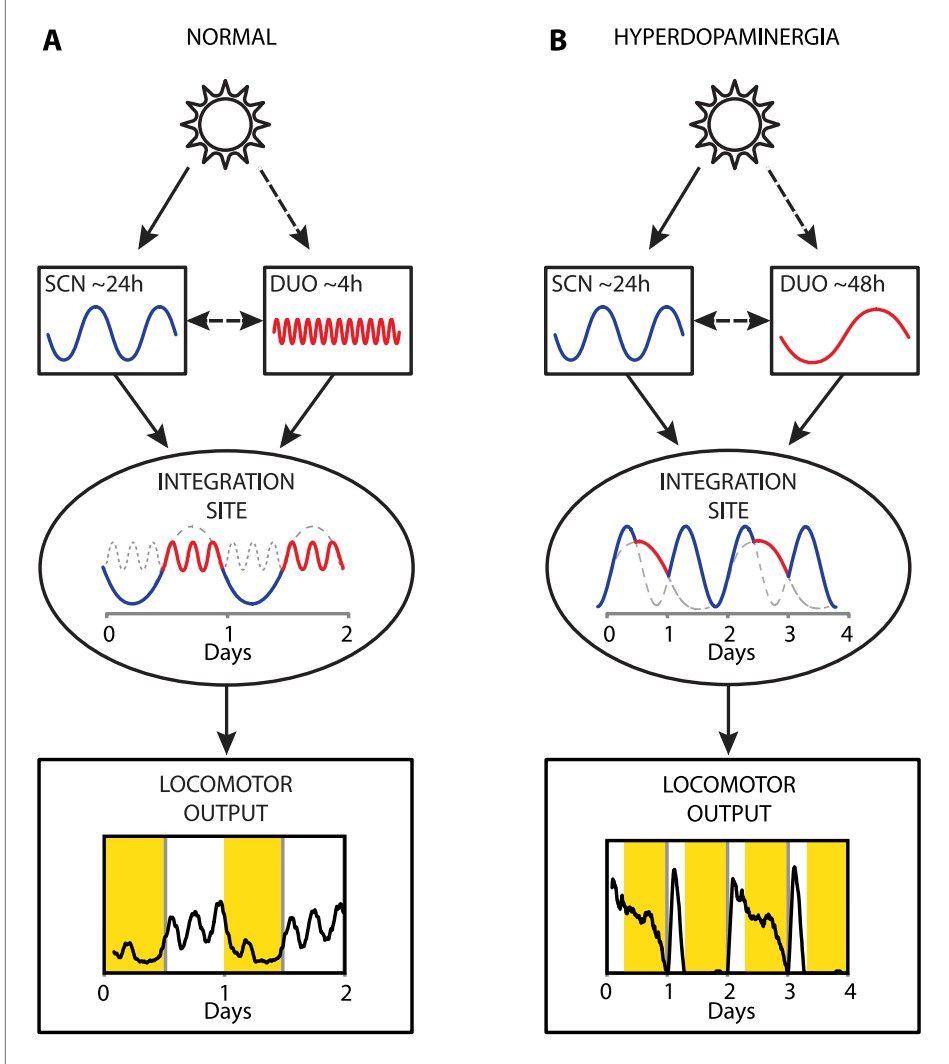

**Figure 9**. Proposed model of circadian and ultradian oscillator output integration to govern daily locomotor behavior. (**A**) Light input entrains the circadian (SCN) and (indirectly) the ultradian (DUO) oscillators creating a stable phase relationship. Their rhythmic outputs, upon integration at a common downstream effector, generate the daily pattern of locomotor activity. (**B**) Under conditions of high DA tone (e.g., DAT elimination or Meth-treatment), DUO period lengthens, leading to a second, separate activity rhythm. This rhythm either free-runs (see *Figure 7D,G*) or phase locks with the SCN pacemaker by adopting a subharmonic, that is, 48-hr period, as frequently observed upon Meth treatment. Representative output plots show average activity of individual mice computed from 8 days of ambulatory activity (**A**) or 14 days of running wheel activity under Meth treatment (**B**).

in *Slc6a3*−/− mice, they repeatedly exhibited an additional activity component with periods longer than 24 hr (*Figure 7D–F*). This component, which was more evident after the first few weeks in DD, extended from the main activity component and lasted for several cycles before disappearing late in the subjective day (*Figure 7D*). Overall, there was no significant difference in the total amount of daily activity even though the temporal pattern of locomotion in these mice is severely altered (*Figure 7C*). The observed dual component activity pattern shows striking similarities to the activity patterns of Meth-treated C57BL/6 mice (*Figure 7G–I*), suggesting that the second rhythmic component is generated by the dopaminergic oscillator.

However, such locomotor activity pattern, with two components oscillating at different frequencies, has also been observed in rats housed under a 22-hr LD cycle and have been attributed to a dissociation of molecular rhythms between the dorsal and ventral SCN (*de la Iglesia et al., 2004*). To rule out that the SCN is similarly affected in *Slc6a3*−/− mice, we sacrificed mice at the peak and trough of the

main activity component when the second rhythm was maximally antiphasic (*Figure 7—figure supplement 1A*). In situ hybridization directed against *Period1* transcripts did not indicate rhythmic desynchrony between or within SCN hemispheres. The riboprobe showed strong, homogenous staining throughout the SCN in the early subjective day, whereas little to no signal was detectable in the early subjective night, suggesting that SCN pacemaker function is unperturbed in *Slc6a3⁻/⁻* mice (*Figure 7—figure supplement 1B*).

Together, these findings thus argue that the second rhythmic component of *Slc6a3⁻/⁻* mice is indeed a product of the above described dopaminergic oscillator.

## Chemogenetic activation of DA neurons lengthens ultradian period

Since striatal and midbrain DA levels vary concordantly with activity rhythms in *Bmal1⁻/⁻* mice, and DA neurons of the SN/VTA are by and large the exclusive source of striatal and midbrain DA, these neurons are a likely site of ultradian rhythm regulation, if not generation. To corroborate the importance of these neurons in the oscillator process, we employed a chemogenetic approach based upon DREADD (designer receptors exclusively activated by designer drugs) technology (*Krashes et al., 2011*). We stereotaxically delivered the adeno-associated virus *AAV-DIO-hM3Dq-mCherry* (*Krashes et al., 2011*) into the SN/VTA region of mice that carried an *Slc6a3* promoter driven *Cre* transgene (*DAT-Cre* ) (*Figure 8A*) (*Turiault et al., 2007*) and that additionally were either BMAL1-deficient (*DAT-Cre, Bmal1⁻/⁻*) or received an SCN lesion (*DAT-Cre*, SCNx). Upon Cre-mediated recombination, virally transfected cells express the stimulatory DREADD hM3Dq, which has been demonstrated to increase neuronal firing frequency upon binding of the compound clozapine-N-oxide (CNO), which is otherwise physiologically inert (*Alexander et al., 2009*). As augmenting firing frequency in DA neurons enhances DA release (*Sulzer, 2011*), and CNO stimulation of successfully transduced *DAT-Cre* mice evokes vigorous firing of dopamine neurons (*Wang et al., 2013*), CNO is expected to increase DA release. Consistently, i.p.-injection of CNO elevated extracellular DA in the striatum of those mice and led to a prolonged increase in locomotor activity (*Figure 8C,D*), which was absent in vehicle injected animals (*Figure 8—figure supplement 1D*). Responsive mice showed mCherry fusion tag expression selectively in tyrosine hydroxylase (TH) positive cells (*Figure 8B*, *Figure 8—figure supplement 1A*). Cell counting revealed that, 92.9 ± 5.1% and 87.8 ± 6.0% (mean ± SEM, N = 6 for both groups) of the midbrain TH⁺ neurons also expressed the mCherry fusion tag in responsive *DAT-Cre*, SCNX and *DAT-Cre, Bmal⁻/⁻* mice, respectively, confirming effective Cre-mediated hM3Dq expression in midbrain DA neurons. Running wheel activity monitoring of virus-injected mice showed the expected short period locomotor oscillations, however, upon switching to CNO-containing drinking water (red arrows), both, *AAV-hM3Dq* transduced *DAT-Cre*, SCNx and *DAT-Cre, Bmal1⁻/⁻* mice responded with locomotor period lengthening (*Figure 8E–H*), an effect that was reversed when mice were returned to pure water (*Figure 8G*, blue arrows). As expected, a period lengthening was not observable in *AAV-hM3Dq* injected *DAT-Cre⁰/⁰*, *Bmal1⁻/⁻* mice upon CNO exposure (*Figure 8—figure supplement 1B,C*). These results further corroborate that elevating DA tone, in this case by selective excitation of DA neurons, lengthens ultradian locomotor period and that the oscillator driving ultradian rhythmicity comprises DA neurons of the VTA/SN region.

## Discussion

Collectively, our results provide strong evidence that a dopaminergic ultradian oscillator (DUO) driving rhythms of behavioral arousal is continuously operative in the mammalian brain. We propose that under normal conditions, this DUO cycles in harmony with the circadian SCN pacemaker and that the rhythmic information of both the SCN and the DUO are integrated at a common downstream site to create the daily pattern in locomotor activity (*Figure 9A*). However, elevation of DA tone can lead to DUO period lengthening, which either results in DUO free-run (for example, *Figure 7D,G*) or reinstatement of oscillator synchrony albeit at a different harmonic (*Figure 9B*, e.g., 48 hr). The DUO appears as a highly tunable oscillator, able to adopt period lengths from a few hours to multiple days. This is in stark contrast to the circadian timer which cannot adopt periods that are more than a few hours off from 24 hr when its limits of entrainment are tested experimentally (*Aschoff and Pohl, 1978*).

Given that DA is known to stimulate locomotor activity (*Zhou and Palmiter, 1995*), our observation of a cyclical rise in extracellular striatal DA, which is in synchrony with ultradian activity rhythms, argues that DA acts as an output of the DUO. Our finding that manipulations affecting extracellular DA levels alter oscillator period and that extracellular DA tone shows a remarkably high correlation with activity

cycle length, strongly suggests that DA is a period determinant and therefore must be an integral component of the oscillatory mechanism itself. As all period-altering manipulations directly impinge upon DA neuronal physiology, this suggests that either (i) DA neurons are the site of ultradian rhythm generation or (ii) they are a key cog in the oscillatory mechanism. The chemogenetic activation experiments indicate that the relevant DA neuronal population is located in the midbrain as selective activation of DA neurons in this region had a period lengthening effect on the ultradian activity. Future experiments will be aimed at delineating the precise DA population(s) required for the ultradian rhythm generation process and how rhythmic synchrony between neurons is maintained, if the DUO is indeed composed of a population of cellular oscillators.

Our data indicate that the circadian and ultradian locomotor rhythms are normally harmonized (for instance, *Figure 9A*), suggesting that the circadian pacemaker and the DUO interact. Of note, extracellular DA was reported to fluctuate diurnally in the rodent striatum (*Hood et al., 2010*), which was abrogated in *Slc6a3⁻/⁻* mice (*Gallardo et al., 2014*). In these studies, only group averaged DA profiles were presented and DA was solely measured in circadian intact mice and rats, thus it is conceivable that any ultradian component (in WT) or infradian component (in *Slc6a3⁻/⁻*) escaped detection. Critically, the observation that the DA levels, on average, followed a diurnal pattern with a night-time peak, together with the observation that ultradian locomotor period in female hamsters is longer in the dark vs the daily light phase (*Prendergast et al., 2012*), is consistent with the LD cycle and/or the circadian pacemaker affecting DUO rhythmicity by altering (extracellular) DA. Note however, that dopamine content in whole brain extracts from circadian-intact rats kept under LD showed a strong ultradian variation with no obvious evidence for a diurnal rhythmic component (*Scheving et al., 1968*). Also, microdialysis did not reveal a day:night difference in extracellular DA levels in the striatal NAcc region when measured in DD (*Chung et al., 2014*). This same study reported elevated DA levels and hyperactivity in mice lacking clock gene *Rev-erbα*, suggesting that the circadian clock, possibly intrinsic to DA neurons themselves, has a role in DA regulation. Indeed, knockdown of the core circadian clock component *Clock* in DA neurons of the murine VTA, increases electrical firing rate in VTA neurons and enhances locomotor response to novel objects (*Mukherjee et al., 2010*). Interestingly, we did not observe any systematic differences in ultradian locomotor period between SCNx and *Bmal1⁻/⁻* mice in constant darkness conditions, regardless of DAT status (*Figures 1, 2*), suggesting that extra-SCN circadian clocks have no role in DUO-mediated ultradian locomotor rhythm generation.

Our data also provide evidence that the postulated MASCO (*Tataroglu et al., 2006*) represents a long-period manifestation of the DUO as a result of methamphetamine's action on the dopamine transporter, blocking dopamine reuptake thereby increasing extracellular DA levels. Interestingly, the dopamine system has been also implicated with another behavioral timing system: the food-entrainable oscillator (FEO). This circadian independent oscillator (*Pitts et al., 2003*; *Pendergast et al., 2009*; *Storch and Weitz, 2009*) drives food-anticipatory locomotor activity (FAA) that emerges when food access is restricted to a few hours each day (*Mistlberger, 2011*). D1 (DRD1) and D2 (DRD2) receptor antagonists attenuate FAA additively (*Liu et al., 2012*) while pharmacological DRD2, but not DRD1, activation altered the phase of FAA (*Smit et al., 2013*). Most recently, using knockout mice, it was revealed that DRD1 but not DRD2 is necessary for the appropriate expression of FAA and that rescuing dopamine signaling selectively within the dorsal striatum was sufficient to restore FAA in dopamine-deficient mice (*Gallardo et al., 2014*). Given the links to the dopaminergic system, it will be of interest to investigate whether the DUO has a role in the temporal regulation of FAA.

While these findings argue that food cues engage the dopamine system to alter daily locomotor activity patterns, it is clear that dopamine signaling also affects food intake as mice lacking dopamine are lethargic and do not actively consume food (*Zhou and Palmiter, 1995*). Considering that the DUO is a universal driver of ultradian behavioral rhythms in mammals, the finding that ultradian bouts of running-wheel and feeding activity are co-expressed in the common vole (*van der Veen et al., 2006*) suggests that dopamine can synchronously drive food seeking and general activity, which is in line with the view that dopamine acts as a general promoter of motivated arousal.

*Slc6a3⁻/⁻* mice have been proposed as a model for schizophrenia (*Gainetdinov et al., 2001*) and the DA hypothesis of schizophrenia states that DA elevation is causal to the behavioral symptoms of this psychiatric condition. Intriguingly, circabidian (48 hr) or free-running rhythms in locomotor activity reminiscent of the behavioral patterns we detected in *Slc6a3⁻/⁻* or Meth-treated mice (*Figures 4F, 7C,F, 9B*) have been observed in schizophrenic subjects (*Wirz-Justice et al., 2001*; *Wulff et al., 2012*),

suggesting that DUO dysregulation underlies the rest-activity aberrations associated with schizophrenia. Furthermore, actigraphy recordings in schizophrenic patients also revealed that Hal treatment reduces circadian/diurnal locomotor amplitude and leads to the emergence of ultradian activity bouts (*Wirz-Justice et al., 1997*, *2009*), effects we likewise observed in *Slc6a3⁻/⁻* mice in response to Hal. Transient DUO period lengthening might equally account for the rest-activity pattern abnormalities observed during manic episodes in bipolar disorder, which has been also associated with altered DA tone (*Berk et al., 2007*). Bipolar subjects have been reported to show rapid, 48-hr cycling between mania and depression (*Gann et al., 1993*; *Wilk and Hegerl, 2010*), one to multiple 48-hr sleep–wake cycles when switching from depression to mania (*Wehr et al., 1982*), or a long-period 'free-running' rhythm of wakefulness (*Wehr et al., 1998*). Thus schizophrenic and bipolar subjects both appear to exhibit rest-activity cycle aberrations strikingly similar to those observed in *Slc6a3⁻/⁻* or Meth-treated mice, suggesting that DUO dysregulation is a common indicator for these psychopathologies and perhaps even a common disease cause.

A switch to sleep–wake cycles with a period much longer than 24 hr have also been observed in subjects that were studied in temporal isolation (*Aschoff, 1965*). Because other physiological parameters such as urine secretion and core body temperature showed phase-aligned, circadian fluctuations with a period much closer to 24 hr, the subjects were considered internally desynchronized. Notably, affected subjects tend to exhibit high scores of neuroticism (*Wever, 1979*). It is therefore conceivable that the observed internal desynchronization is also due to a dysregulation of the DUO.

## Materials and methods

### Animals

*Bmal1⁻/⁻* (*Bunger et al., 2000*) and *DAT-Cre* (*Turiault et al., 2007*) mice were on a C57BL/6J genetic background, while *Slc6a3⁻/⁻* mice (*Giros et al., 1996*) were maintained on a mixed C57BL/6JxDBA/2J background (*Morice et al., 2004*). Animals were housed under LD 12 hr:12 hr unless otherwise stated. *Slc6a3⁻/⁻* mice were found to exhibit high attrition rates (~80%) in experiments that involve long-term locomotor activity monitoring. To increase survival, *Slc6a3⁻/⁻* mice were first group-housed for 1 week after transfer into the light-tight, ventilated cabinets used for activity monitoring. This was followed by 1 week of individual housing in running wheel cages with the wheels locked before commencing baseline activity recordings. During this adaptation phase and throughout the subsequent experimental period, both *Slc6a3⁻/⁻* mice and their wild-type littermates were provided with ad libitum chocolate flavored chow (Supreme Mini-Treats, BioServ, Flemington, NJ) as well as cotton nestlets and ample shredded corrugated card stock. This regimen significantly reduced attrition rates to less than 20% on average. 'Material and methods' were performed in accordance with the Canadian Council on Animal Care guidelines and approved by the McGill University Animal Care Committee.

### Locomotor activity and core body temperature monitoring

*Running wheels:* Animals were individually housed in light-controlled cabinets and activity was recorded continuously (ClockLab, Actimetrics, Wilmette, IL). Actograms, displaying binned running wheel revolutions per 6 min (0.1 hr), and the associated Lomb-Scargle periodograms, displaying amplitude, were generated using ClockLab software. *Telemetry:* Animals were individually housed in standard cages placed atop energizer/receiver units (ER-4000, Starr Life Science Corp., Oakmont, PA). 1 week prior to data collection, electromagnetic induction powered telemetry probes (G2 E-mitter, Starr Life Science Corp.) were implanted intraperitoneally. Locomotion, measured in counts per minute, and core body temperature (BT, in °C), was collected in 6-min bins (0.1 hr) using Vitalview software (Starr Life Science Corp.). BT data was exported into Clocklab to generate actogram-style data displays with tick mark height corresponding to temperatures from 34–38°C; for waveform generation, averaging, and 2 hr recursive smoothing, locomotor activity and BT data was exported to Excel (Microsoft, Redmond, WA). Matlab (Mathworks, Natick, MA) was used for ribbon plot generation, and low pass Butterworth filtering (1 hr).

### Pharmacology

Stock solutions of (+)-Methamphetamine hydrochloride (100 mg/l, Sigma-Aldrich, St. Louis, MO), D-Amphetamine hemisulfate (100 mg/l, Sigma-Alrdich), Haloperidol (11.25 mg/ml, Sigma-Aldrich) and Clozapine-N-Oxide (15 mg/l, National Institute of Mental Health, Bethesda, MD) were prepared

using distilled water. Methamphetamine solutions were adjusted to pH 7 using sodium hydroxide and haloperidol was dissolved by stirring at 40°C.

## Chronic methamphetamine infusion

Mice were continuously infused with methamphetamine (0.6 mg/day in 0.9% saline) for 14 days using subcutaneously implanted osmotic minipumps (Alzet Model 1002, Durect Corp., Cupertino CA). Pumps were fitted with a 65-mm polyvinyl chloride catheter (0.69-mm inner diameter, Plastics One, Roanoke, VA) and backfilled with saline to allow for an approximately 4 day post-surgery recovery prior to drug exposure.

## In situ hybridization

In situ hybridization was performed as previously described (*Kraves and Weitz, 2006*). Briefly, following decapitation, brains were removed and quickly frozen at −80°C. Serial coronal hypothalamic brain slices (25 µm) were collected using a cryostat and stored at −80°C until hybridization. Sections were hybridized overnight at 60°C to a digoxigenin-labeled riboprobe targeting the coding regions of mouse *Per1* (nucleotides 579–1478 of the *Per1* mRNA, Genbank, NM_011065.4).

## SCN lesions

Electrolytic lesions of the suprachiasmatic nucleus were performed as described (*Storch et al., 2007*). Briefly, an electrode (RNE-300X, Rhodes Medical Instruments, Woodland Hills, CA) was lowered through a hole drilled in the skull at the mid-sagittal sinus according to stereotaxic coordinates (AP −0.25 mm, DV −6.00 mm from Bregma) and a constant current (2 mA, 10 s; D.C. Constant Lesion Maker, Grass Instruments, Quincy, MA) was applied. Only mice with behavioral circadian arrhythmia (in the 20–28-hr range assessed by Lomb-Scargle periodogram analysis) and subsequent post-mortem histological verification using DAPI staining mounting medium (Vectashield, Vector Labs) were included for analysis. This represented approximately 50% of lesioned mice when averaged across all studies.

## In vivo microdialysis

1 week prior to sampling, mice were stereotaxically implanted with a guide cannula (C312G/spc 2.5 mm below pedestal, Plastics One), targeting the striatum (AP +1.10 mm, DV +5.5 mm, ML +0.9 mm). 1 day prior to dialysis, mice were transferred to a beam break monitor (VersaMax, Omnitech Electronics, Columbus, OH) and tethered using a dummy probe assembly under constant, dim, red light (<5lux). On the day of sampling, in-house recording probes (*Lupinsky et al., 2010*), were connected to a central swivel (Model 72-0000, Harvard Apparatus, Holliston, MA). Artificial cerebrospinal fluid was delivered via syringe pump (Model 403, CMA Microdialysis, Kista, Sweden) at a flow rate of 1.0 µl/min. This procedure was similarly followed to test the effects of CNO in AAV-transduced *DAT-Cre* mice where intraperitoneal injections of 1 mg/kg CNO were given after approximately 1 hr of baseline sampling. Samples were collected every 20 min for 4 hr and immediately admixed with 1 µl of perchloric acid. DA and DOPAC content in the dialysate were determined by high-performance liquid chromatography with electrochemical detection (HPLC-EC) as described (*Domenger et al., 2012*). Chromatographic peak analysis was conducted using CoulArray software (ESA Inc., Chelmsford, MA). After data collection, brains were removed, sectioned, and stained with hematoxylin to verify probe placement. Locomotor activity data, recorded as the number of beam breaks per minute, were exported into Matlab (Mathworks) for waveform generation, 20-min binning, cosinor modeling, and false discovery rate analysis.

## DA/DOPAC content determination in tissue extracts

After decapitation, brains were quickly removed and frozen (−80°C). 320 µm coronal brain slices were obtained by cryosectioning and then microdissected (1- or 2-mm diameter sample corers, Fine Science Tools Inc., Foster City, CA) to obtain tissue of the CPu (2 mm), NAcc (1 mm), and SN/VTA midbrain (2 mm) regions (see *Figure 6G* for punch location). DA/DOPAC was quantified as previously described (*Domenger et al., 2012*). Briefly, individual punches from each region were homogenized in 45 µl perchloric acid (0.25 M) to which 15 µl of a 100 ng/ml solution of 3,4-dihydroxybenzylamine was added, which served as the internal standard. Concentrations were determined by HPLC-EC. After perchloric acid extraction, the protein containing pellets were reconstituted in 0.1 N sodium hydroxide for protein quantification (Pierce BCA Kit, ThermoFisher Scientific, Waltham, MA).

## Chemogenetics

Mice were anaesthetized with isofluorane and placed in a stereotaxic aparatus (David Kopf Instruments). Recombinant *AAV8-DIO-h3MDq-mCherry* (*Krashes et al., 2011*) (titer = $3 \times 10^{12}$ genomes copies per ml, UNC Vector Core Services, Chapel Hill, NC) was bilaterally injected into to the VTA/SN area (coordinates from bregma: AP: −3.44 mm, DV: −4.40 mm, L: ±0.48 mm) (*Tsai et al., 2009*) through a cannula (33 gauge, Plastics One) at a flow rate of 0.05 µl/min for 10 min (0.5 µl total volume per side) using a syringe pump (Harvard Apparatus). Mice were subsequently maintained in individual housing for at least 2 weeks prior to CNO treatment.

## Immunohistochemistry

Immunostaining was performed as previously described (*Chu et al., 2013*) using cryoprotected (30% Sucrose), free-floating coronal sections from fixed brains collected after intra-cardiac perfusion (4% paraformaldehyde) and cut at 40 microns using a cryostat (Leica, Solms, Germany). For fluorescent labeling, sections were incubated overnight with primary antibodies for mCherry (rabbit anti-RFP, 1:1000, Rockland, Limerick, PA) to enhance detection of the mCherry expression and antibodies for tyrosine hydroxylase (mouse anti-TH, 1:1000, EMD Millipore, Etibicoke, Canada). This was followed by 2 hr incubation with secondary antibodies (anti-rabbit Alexa Fluor 567 and anti-mouse Alexa Fluor 647, 1:250, Life Technologies, Carlsbad, CA) after which sections were mounted on superfrost slides (VWR, Radnor, PA), coverslipped with Vectashield mounting medium (Vector Labs, Burlington, Canada) and imaged by fluorescence microscopy (AxioObserver Z1, Zeiss, Jena, Germany). Quantification of co-expression was performed on a single, medial, midbrain section from each behaviorally responsive mouse using the cell counter plugin of ImageJ (National Institute of Health, Bethesda, MD) in order to determine the percentage of all TH-positive cells that also expressed the hM3Dq--mCherry fusion protein (N = 6 per genotype).

## Statistical analysis

One-way and repeated measures ANOVA, t-tests, and linear regression analyses were performed using Prism 5 (GraphPad, La Jolla, CA). Planned comparisons for the repeated measures were carried out to determine significant period changes between subsequent measurements. Bonferroni correction was used for post-hoc testing of individual group differences.

## False discovery rate (FDR) and cosinor determination

To evaluate the probabilistic significance of the least-square fit between the DA trace and the cosinor model that was computed based on the animal's locomotor activity oscillations, a false discovery rate approach was employed (*Storch et al., 2007*). For cosinor computation, we first determined the locomotor period of *Bmal1*$^{−/−}$ mice by Lomb-Scargle periodogram analysis of 20 hr of locomotor recordings following dialysate sampling. The determined period was then used as an input parameter for a least-squares cosinor analysis (*Nelson et al., 1979*) using a custom script and function written for Matlab (https://github.com/storchlab/Cosinor-FDR.git). This procedure generated a best-fit (co)sine wave modeling the activity time-series. This model wave was then used to assess the concordance between the extracellular DA fluctuations and locomotor oscillations. To calculate FDR, we randomly permutated the temporal order of the measured DA concentrations 100,000 times, and assessed its fit to the cosinor. The permutation procedure will degrade the fit of any truly rhythmic signal but will have little to no effect on noisy or flat profiles. The probability of false discovery was calculated as the percentage of randomly permuted traces that show an equal or better fit to the cosinor than the measured DA fluctuation.

## Butterworth filtering

Low pass filtering (1-hr cut-off) of raw data was conducted using a Butterworth zero-phase filter (*Butterworth, 1930*) in Matlab. This allowed for better visualization of ultradian frequencies in the >1-hr range, in a manner similar to recursive smoothing but without phase distortion.

## Period determination

Ultradian rhythms show variability in amplitude and period. We thus used Lomb-Scargle periodogram analysis (Clocklab) to estimate ultradian period length as this method is relatively tolerant to noisy data and data gaps, which can result from intermittent ultradian rhythm expression (*Press and Rybicki, 1989*). Unless otherwise stated, period length was determined by identifying the highest peak above the significance threshold (α = 0.01) in the Lomb-Scargle periodogram. Subsequent

plotting of an actogram at the determined modulus was conducted in each case in order to visually confirm rhythmicity and rule out the false identification of harmonic frequencies or side-lobes due to leakage (*Van Dongen et al., 1999*).

## Continuous wavelet transforms

To visualize and quantify the dynamics of period and amplitude in circadian incompetent mice, continuous wavelet transforms using the Morlet wavelet (*Goupillaud et al., 1984*) were performed for the 1–12-hr range using custom algorithms written in Matlab (https://github.com/storchlab/CWT.git). Locomotor activity data of seven consecutive days per individual SCN-lesioned or $Bmal1^{-/-}$ mouse was used. For each spectrogram, a ridge identifying the peak amplitude of the spectrum was generated as described (*Leise, 2013*) and the associated values were used to calculate the average and standard deviation of the dominant ultradian frequency for each animal.

## Amplitude spectral density (ASD)

To determine the prevalence of oscillations in a given period range we calculated the area under the curve of all significantly rhythmic periodicities ($\alpha = 0.01$) in the Lomb-Scargle periodogram, that is, the spectral density. For normalization, the obtained spectral density was divided by the total significant spectral density in the 0–40-hr range and expressed as percentage.

## Acknowledgements

We thank A Villemain, E Vigneault, and M-E Desaulnier for mouse-colony maintenance and F Tronche for the *DAT-Cre* mice. This work was supported by a NSERC discovery grant (K-FS), CIHR operating grants (K-FS and AG), a CIHR New Investigator Award (K-FS), by the Graham Boeckh Foundation (BG) and a Canada Research Chair (BG).

## Additional information

### Funding

| Funder | Grant reference number | Author |
| --- | --- | --- |
| Canadian Institutes of Health Research | Operating Grants | Alain Gratton, Kai-Florian Storch |
| Natural Sciences and Engineering Research Council of Canada | Discovery Grant | Kai-Florian Storch |
| Graham Boeckh Foundation | Graham Boeckh Chair in Schizophrenia | Bruno Giros |
| Canadian Institutes of Health Research | New Investigator Award | Kai-Florian Storch |
| Canadian Institutes of Health Research | Canada Research Chair | Bruno Giros |

The funders had no role in study design, data collection and interpretation, or the decision to submit the work for publication.

### Author contributions

IDB, K-FS, Conception and design, Acquisition of data, Analysis and interpretation of data, Drafting or revising the article; LZ, LM, Acquisition of data, Analysis and interpretation of data; MVK, AG, BG, Conception and design, Drafting or revising the article, Contributed unpublished essential data or reagents

### Author ORCIDs

Ian D Blum, http://orcid.org/0000-0001-6999-8364

### Ethics

Animal experimentation: All experimental procedures were performed in accordance with the Canadian Council on Animal Care guidelines and approved by the McGill University Animal Care Committee (animal use protocol #2010-5945).

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
