## [Decision Letter]

Thank you for sending your work entitled “A dopaminergic oscillator generates ultradian rhythms of behavioral arousal” for consideration at *eLife*. Your article has been favorably evaluated by a Senior editor and 3 reviewers, one of whom is a member of our Board of Reviewing Editors.

All the individuals responsible for the peer review of your submission have agreed to reveal their identity: Richard Palmiter (Reviewing editor), Joseph Takahashi, and Andrew Steele (peer reviewers).

Because all three reviewers were impressed by the quality of the work presented and all agreed that the paper should be accepted there was no need for discussion among the reviewers. No additional experiments are required. The results are compelling, but there are a few loose ends to consider:

1) Manipulations of DA signaling will have many effects on physiology and behavior including food-entrained oscillations in activity. Thus, it seems important to know how the various DA signaling manipulations affect patterns of food intake. Acknowledgment of the potential interactions between feeding behavior and locomotion would be welcome.

2) It would be useful to include histograms comparing the overall locomotor activity of the comparison groups in addition to the histograms showing average period length. Why do DAT KO mice have lower activity on a running wheel compared to WT mice (Figure 6), but apparently more activity in their home cage (Figure 3)?

3) It seems most likely that DA is a “key cog in the oscillatory mechanism” rather than the generator of the oscillations because mice lacking DA can have robust bouts of activity. For example, mice lacking DA are, indeed, hypoactive and they become extremely hyperactive in response to drugs that restore DA, but then paradoxically they have another bout of DA-independent activity 24-30 hours later. When this experiment is performed in obese mice (leptin deficient), there can be several very robust bouts of DA-independent activity that appear to be circadian (Sczcypka et al., 1999, PNAS 96, 12138; Szczypka et al., 2000, Nat Genet 25, 102). These results suggest that bouts of activity can be entrained by surges in DA signaling but can be maintained without DA; thus, some other circuit must maintain the rhythms.

---

## [Author Response]

In addition to the revisions in response to the reviewers’ comments, we would also like to propose the following additional changes to the manuscript, if you agree with it.

We changed the title to “A highly tunable dopaminergic oscillator generates ultradian rhythms of behavioral arousal” as we think it better reflects the nature of the oscillator system we describe (we also adjusted the main text at a few places accordingly). This request for a title change is particularly owed to our finding that the oscillator is capable to adopt period lengths from a few hours to multiple days by changing DA tone.

We now added running wheel data from *Bmal1*^*-/-*^*,Slc6a3*^*-/-*^ double KO mice and a respective bar graph reporting their ultradian periods. As these mice largely phenocopy the SCNx-*Slc6a3*^*-/-*^ animals, these data critically strengthen our core finding which is ultradian period lengthening by *Slc6a3* disruption. It also provides additional support to our claim that extra-SCN clocks have no role in the ultradian rhythm generation process and that this process is not affected by the developmental absence of BMAL1 and thus clock function.

To accommodate the new data we split Figure 2, extending the total Figure count to 9.

*1) Manipulations of DA signaling will have many effects on physiology and behavior including food-entrained oscillations in activity. Thus, it seems important to know how the various DA signaling manipulations affect patterns of food intake. Acknowledgment of the potential interactions between feeding behavior and locomotion would be welcome*.

While we have not explicitly explored the temporal feeding pattern in our experimental models: there is data from the common vole demonstrating that ultradian feeding and locomotor activity rhythms are co-expressed, suggesting that the DUO (and thus dopamine) drives both synchronously. We have extended the Discussion section accordingly.

*2) It would be useful to include histograms comparing the overall locomotor activity of the comparison groups in addition to the histograms showing average period length. Why do DAT KO mice have lower activity on a running wheel compared to WT mice (*Figure 6*), but apparently more activity in their home cage (*Figure 3*)*?

While we appreciate that it might appear as though overall locomotion differs between groups this is not the case. Analysis of the datasets in question revealed that total daily activity was not significantly different in either the telemetry or the RW. These results have now been added (Figures 4 and 7). The erroneous impression of genotypic differences in activity levels is likely due to the fact that *Slc6a3*^*-/-*^ mice typically exhibit fewer but longer and more consolidated (less fragmented) activity bouts, leading to higher peak activity levels (in Figure 4). Similarly, these bout-character differences likely account also for the erroneous impression of differences in daily running wheel activity (Figure 7). Together, the new data supports our original analysis in that it is the pattern of locomotion, which is substantially altered by DAT removal, whereas the total daily activity is not.

*3) It seems most likely that DA is a “key cog in the oscillatory mechanism” rather than the generator of the oscillations because mice lacking DA can have robust bouts of activity. For example, mice lacking DA are, indeed, hypoactive and they become extremely hyperactive in response to drugs that restore DA, but then paradoxically they have another bout of DA-independent activity 24-30 hours later. When this experiment is performed in obese mice (leptin deficient), there can be several very robust bouts of DA-independent activity that appear to be circadian (Sczcypka et al., 1999, PNAS 96, 12138; Szczypka et al., 2000, Nat Genet 25, 102). These results suggest that bouts of activity can be entrained by surges in DA signaling but can be maintained without DA; thus, some other circuit must maintain the rhythms*.

We are very aware of the very intriguing findings in Szczypka et al. (1999). However, we do not think that the results presented in this study (and the Szczypka et al., 2000 study) particularly refute the hypothesis that DA acts as a key element of the ultradian rhythm generation process. As the DD mice examined in these studies were circadian competent and given that the DA-independent activity bout emerged ∼24hrs after the previous bout, it seems plausible that this DA-independent bout is driven by the circadian clock in the SCN and/or other sites (possibly involving input to midbrain DA neurons, other ascending arousal nuclei, or striatal post-synaptic neurons).

We actually wish to critically test this, as a next step towards deciphering the DUO mechanism, by performing SCN lesion in DD mice and attempting to recapitulate the phenomena observed in Szczypka et al. Thus, the current state of knowledge suggests that DA acts as the key mediator of DUO action (in addition to being a period determinant), while other (circadian) cues are able to engage arousal/locomotion circuitry in a DA/DUO independent fashion (this view is e.g. also supported by the presence of a daily, night-time activity bout in addition to the circadian day-time activity bout in Meth-treated, wild-type mice; see actogram in Figure 9).